# Waist Circumference and Abdominal Volume Index Can Predict Metabolic Syndrome in Adolescents, but only When the Criteria of the International Diabetes Federation are Employed for the Diagnosis

**DOI:** 10.3390/nu11061370

**Published:** 2019-06-18

**Authors:** Javier S. Perona, Jacqueline Schmidt-RioValle, Ángel Fernández-Aparicio, María Correa-Rodríguez, Robinson Ramírez-Vélez, Emilio González-Jiménez

**Affiliations:** 1Instituto de la Grasa-CSIC, Campus Universidad Pablo de Olavide, Edificio 46, 41013 Seville, Spain; perona@ig.csic.es; 2Department of Nursing, University of Granada, Av. Ilustración, 60, 18016 Granada, Spain; anfeapa@ugr.es (Á.F.-A.); macoro@ugr.es (M.C.-R.); emigoji@ugr.es (E.G.-J.); 3Department of Health Sciences, Public University of Navarra, Navarrabiomed- IdiSNA, Pamplona, 31006 Navarra, Spain; robin640@hotmail.com

**Keywords:** anthropometric indexes, diagnosis criteria, metabolic syndrome, adolescents, obesity

## Abstract

We previously reported, using the diagnostic criteria of the International Diabetes Federation (IDF), that waist circumference (WC) and abdominal volume index (AVI) were capable of predicting metabolic syndrome (MetS) in adolescents. This study was aimed at confirming this finding when other diagnostic criteria are used. A cross-sectional study was performed on 981 Spanish adolescents (13.2 ± 1.2 years). MetS was diagnosed by eight different criteria. Ten anthropometric indexes were calculated and receiver-operator curves (ROC) were created to determine their discriminatory capacity for MetS. Of all diagnostic criteria, the ones proposed by the IDF showed the highest mean values for weight, WC and systolic blood pressure in boys and girls with MetS, and the lowest for glucose and triglycerides in boys. ROC analysis showed that only WC, AVI and body roundness index (BRI) achieved area under the curve (AUC) values above 0.8 in boys, and that fat content, body mass index (BMI), WC, AVI, BRI and pediatric body adiposity index (BAIp) showed AUC values above 0.8 in girls. Importantly, this occurred only when diagnosis was carried out using the IDF criteria. We confirm that WC and AVI can predict MetS in adolescents but only when the IDF’s diagnostic criteria are employed.

## 1. Introduction

Metabolic syndrome (MetS) is defined as the clustering of risk factors for cardiovascular disease (CVD) and type 2 diabetes mellitus, such as hypertension, central obesity, atherogenic dyslipidemia, and insulin resistance [1]. It is now recognized that MetS is not only a problem of adulthood. Children that suffer changes in their metabolic profile present a higher risk of developing this condition in early stages of their lives, like adolescence [2,3], with the resulting risk of developing type 2 diabetes mellitus and cardiovascular disease [4].

For this reason, it is necessary to identify early changes in the metabolic profile for the diagnosis of this condition in children and adolescents [5]. In addition, scientific evidence shows that the use of anthropometric indexes is a simple and innocuous method that can discriminate regional fat to predict disorders as MetS in adolescents. In a previous study with 981 Spanish adolescents, we concluded that waist circumference (WC) and abdominal volume index (AVI) are the anthropometric indexes that best predict MetS in this population Perona et al. [6]. In that study, however, only the criteria of the International Diabetes Federation (IDF) for the diagnosis of MetS [7] were considered. 

There is currently no consensus over the most effective criterion for the diagnosis of MetS in the adolescent population [8,9], in all likelihood because of the different sets of components used in the different diagnostic criteria [10]. In recent years, adaptations of the criteria established for the diagnosis of MetS in adults, such as the National Cholesterol Education Program—Adult Treatment Panel III (NCEP–ATP III) [11], modified by Cook et al. [12], Weiss et al. [13], Duncan et al. [14], and de Ferranti et al. [15], have been used. Unlike others, the criteria established by the IDF [7] consider the presence of abdominal obesity as a pre-requisite for the diagnosis of MetS. In contrast, Cook et al. [12], Cruz and Goran [16], de Ferranti et al. [15], and Rodríguez et al. [17] consider the presence of three or more impaired components for the diagnosis, even if adolescents do not have abdominal obesity. Viner et al. [18], following modified WHO criteria adapted for children (Alberti and Zimmet [19]), considered the presence of at least four impaired components as necessary for the diagnosis of MetS. 

This variability in the criteria leads to striking differences in the diagnosis of MetS and might affect the results of our previous study. Therefore, the present study is aimed at verifying whether WC and AVI still maintain their predictive capacity for MetS in adolescents when other criteria, such as those aforementioned, are used together with the ones proposed by the IDF. 

## 2. Materials and Methods

### 2.1. Study Design and Participants

This cross-sectional study included 981 adolescents (456 boys and 525 girls) with a mean age of 13.2 ± 1.2 years (11–16 years old). All of the participants had been born in Spain and also resided there. They came from families of a similar socio-economic level and attended 18 educational centers in the provinces of Granada and Almeria (southeastern Spain). Ten of these schools were public, and eight were private. The principals were sent a letter inviting their schools to participate in the study. In all cases, the invitation was accepted. In each school, two classes of a total of three per grade were randomly selected for participation. The flow diagram in Figure 1 shows the recruiting process. The Education Boards of Granada and Almeria gave their approval to the research study before it was carried out. The study also received the authorization of the principals of the participating school.

The Ethics Committee of the University of Granada approved both the study and the model of informed consent used. The parents and legal guardians of the participants gave their written informed consent at the beginning of the study. In addition, the study complied with the International Code of Medical Ethics of the World Medical Association and the Helsinki Declaration. To be included in the study, students had to be healthy and not suffer from any endocrine or physical disorder.

### 2.2. Anthropometric Measurements

All participants underwent an anthropometric evaluation, which was performed by a member of the research team, trained for that purpose. The evaluation was carried out according to the criteria established by the International Society for the Advancement of Kinanthropometry (ISAK) [20]. All body measurements were taken in the morning after a 12-h fasting period and 24-h without any physical exercise. The body weight of each participant was measured twice on a self-calibrating SECA 861 class (III) digital floor scale (Saint Paul, MI, USA) with a precision of up to 100 g. All participants were asked to wear light clothes and to remove their shoes beforehand.

Height measurements were taken with a SECA anthropometer (Model 214). Participants without shoes were measured in an upright position with their back and heels in permanent contact with the vertical height rod of the anthropometer and their head oriented in the Frankfurt plane. The horizontal headpiece was then placed on top of their heads. In all cases, the participants’ height and weight were measured twice. The final value was the average of the two measurements.

In addition, the BMI was calculated as the participant’s weight divided by the square of his/her height (kg/m^2^). The WC and hip circumference were measured with a SECA flexible, inextensible measuring tape with an accuracy of 1mm. The WC was measured on a horizontal plane at a point that was equidistant from the lowest floating rib and the upper border of the iliac crest. In all cases, this measurement was taken after exhalation.

Hip circumference was also measured on the horizontal plane and at the maximum protuberance of the buttocks, which coincides in the front with the ischiopubic symphysis. The waist-to-hip ratio (WHR) was calculated by dividing the waist perimeter by the hip perimeter. Also measured were the triceps, biceps, subscapular, and suprailiac skinfolds. The instrument used for this purpose was a Holtain skinfold caliper (Holtain Ltd., Crymych, UK), with an accuracy of 0.1–0.2 mm. The percentage of body fat was based on these skinfold measurements.

Previously, the Brook equation was used to calculate body density [21]. Once the body density value had been obtained, the body fat percentage was determined with the Siri equation [22]. In each educational center, all anthropometric measurements were taken in a classroom that had been especially prepared for this purpose. The privacy of the students was thus guaranteed.

The rest of the anthropometric indexes, abdominal volume index (AVI), body roundness index (BRI), body adiposity index (BAI), body adiposity index for pediatrics (BAIp), conicity index (C-Index), and body shape index (ABSI), were calculated using the following Equations [23,24,25,26,27,28]:
AVI = (2Waist Circumference^2^ (cm) + 0.7(Waist Circumference − Hip Circumference)^2^ (cm))/1000
BRI = 364.2 − 365.5 [1 − π^−2^ Waist Circumference^2^ (m) Height^−2^ (m)]^1/2^
BAI = [Hip circumference (m)/Height^2/3^ (m)] − 18
BAIp = Hip circumference (cm)/Height (m)^0.8^ − 38 
C-Index = 0.109^−1^ Waist Circumference (m) [Weight (kg)/Height (m)]^−1/2^
ABSI = WC (m)/(BMI^2/3^(kg/m2)Height^1/2^ (m))

### 2.3. Serum Biochemical Examination

Blood collection was performed after a 12-h fast. At 8:00 a.m., a nurse member of the research team extracted 10 mL of blood from the median cubital vein of the right arm. For this purpose, a vacutainer system was used with a vacuum blood collection tube. Once the blood had been collected, the glucose concentration was measured with an enzymatic colorimetric method (glucose oxidase-phenol aminophenazone (GOD-PAP); Human Diagnostics, Germany). Also measured were concentrations of HDL-C, total cholesterol, and triglycerides by means of enzymatic colorimetric methods. This was done with an Olympus analyzer.

Four hours after the blood extraction, the samples were centrifuged at 1300 g for 15 minutes (Z400 K, Hermle, Wehingen, Germany). This process separated the red blood cells from the serum, which was then frozen at −80 °C for subsequent analysis. The estimation of low-density-lipoprotein cholesterol (LDL-C) was obtained with the Friedewald equation:LDL-C = Total Cholesterol − HDL-C − (TG/5)
where TG = concentration of triglycerides. Serum insulin was determined by radioimmunoanalysis (Insulin Kit; DPC, Los Angeles, EEUU). Insulin resistance was quantified by Homeostatic Model Assessment (HOMA) [29] with the following equation: fasting glucose (mmol/L) × fasting insulin (mU/L)/22.5.

### 2.4. Blood Pressure Determination

Blood pressure levels were measured by a calibrated aneroid sphygmomanometer and a Littmann^®^ stethoscope (Saint Paul, MI, USA), according to most widely accepted international recommendations [30]. Systolic blood pressure (SBP) ≥130 and/or diastolic blood pressure (DBP) ≥85 mm Hg were considered to be a risk factor of MetS.

### 2.5. Diagnostic Criteria of Metabolic Syndrome

Eight different criteria were used to diagnose MetS in the adolescent sample studied: Cook et al. [12], Weiss et al. [13], Duncan et al. [14], de Ferranti et al. [15], Cruz and Goran [16], Rodríguez-Moran et al. [17], and Viner et al. [18], as well as the IDF criteria as published by Zimmet et al. [7]. The details of the criteria employed may be consulted in Table 1.

### 2.6. Statistical Analysis

The normality of the distribution was assessed using the Kolmogorov-Smirnov test. Results were reported as mean ± SD, except for the number of girls and boys with or without MetS, which was expressed as a number. Student’s t-test was used to assess mean differences between boys and girls. The area under receiver operating characteristic (ROC) curves was calculated to evaluate the abilities of the anthropometric indices to predict MetS. Cutoff points were proposed after calculation of the Youden’s Index (sensitivity+specificity-1). The areas under the ROC curves were compared using DeLong et al.’s [31] non-parametric approach. Based on the assumption that abdominal obesity is a component of MetS, we conducted a multicollinearity test for the anthropometric indexes that included WC (AVI, BRI and ABSI), and the variance inflation factor (VIF) was calculated. Comparisons of means were assessed by ANOVA, followed by Tukey’s test. SPSS v24.0 (IBM, Armonk, NY, USA) was used to perform statistical analyses. Statistical significance was defined as *p* < 0.05.

## 3. Results

### 3.1. Baseline Characteristics of the Participants

Table 2 shows the characteristics of the participants, including anthropometric and biochemical measures, including HOMA-IR. On average, variables were within normal limits, and for most of them, no differences were observed between boys and girls. However, girls showed significantly lower body weight, WC and SBP and higher fat content.

Table 3 and Table 4 show the characteristics of boys and girls, respectively, diagnosed of MetS according to the different diagnostic criteria studied. The number of diagnosed participants varied with the criteria from 25 to 68 boys. In these subjects, the criteria proposed by Viner et al. [18] resulted in 25 individuals with MetS, while those proposed by Duncan et al. [14], Rodriguez-Moran et al. [17] and Cruz and Goran [16] resulted in 68 individuals. Since these latter subjects were actually the same, the mean values of all variables studied (weight, fat %, BMI, WC, fasting glucose, triglycerides, cholesterol, LDL-cholesterol, HDL-cholesterol, SBP, DBP, insulin and HOMA-IR) were also the same. The criteria proposed by the IDF consistently showed the highest mean values for weight, WC, HDL-cholesterol, SBP, DBP and insulin and the lowest for glucose, triglycerides, cholesterol and LDL-cholesterol. In contrast, using the criteria proposed by Viner et al. [18], resulted in the highest mean levels of glucose, triglycerides, cholesterol, LDL-cholesterol and HOMA-IR. Among the components of the MetS, the highest mean values for WC, HDL-cholesterol and SBP were 86.0 cm, 35.7 mg/dL and 137.2 mmHg, respectively, (IDF) and for glucose and triglycerides, 193.7 mg/dL and 338.9 mg/dL, respectively [18]. The lowest mean values for WC, HDL-cholesterol and SBP were 74.6 cm (Cook et al. [12]), 32.4 mg/dL (Viner et al. [18]) and 121.6 mmHg (Cook et al. [12]), respectively. No significant differences were observed in age, fat content and BMI among adolescent boys with MetS, regardless of the criteria used.

In girls, the variability in the number of diagnosed individuals was much higher. The criteria proposed by Weiss et al. [13] resulted in the diagnostic of only 18 individuals with MetS, while those proposed by Cruz and Goran [16] resulted in 171 individuals. The criteria proposed by the IDF consistently showed the highest mean values for weight, BMI, WC and SBP, while the lowest were found when the criteria by Cook et al. [12] were employed. Regarding mean biochemical parameters (glucose, triglycerides, cholesterol and LDL-cholesterol), the highest values were observed in individuals diagnosed using the criteria by Weiss et al. [13], while the lowest corresponded to individuals diagnosed using the criteria by Cruz and Goran [16]. Among the components of the MetS, the highest mean values for WC and SBP were 83.2 cm, and 132.3 mmHg, respectively (IDF), for HDL-cholesterol was 38.0 mg/dL (Cruz and Goran [16]) and for glucose and triglycerides, 194.4 mg/dL and 329.8 mg/dL, respectively (Weiss et al. [13]). The lowest mean values for WC and SBP were 70.5 cm and 113.4 mmHg, respectively (Cook et al. [12]), for HDL-cholesterol was 32.8 (Weiss et al. [13]) and for glucose and triglycerides, 104.5 mg/dL and 145.5 mg/dL, respectively (Cruz and Goran [16]). No significant differences were observed in age, fat content and DBP among girls with MetS regardless the criteria used.

### 3.2. Area under the Curve Values of the Anthropometric Indexes for the Diagnosis of Metabolic Syndrome

Table 5 and Table 6 show the area under the curve (AUC) values obtained from ROC analyses of the different anthropometric indexes for predicting MetS in adolescent boys and girls, respectively, according to the different diagnostic criteria employed. In boys, only WC, AVI and BRI achieved AUC values above 0.8, thus showing a high predictive capacity. These values were obtained when the diagnosis of MetS was performed using the IDF criteria. When using any of the other criteria studied, AUC values were always below 0.8. The lowest AUC values observed, i.e., the lowest predictive capacity of all of the anthropometric indexes, were found when the criteria used for the diagnosis were those proposed by Weiss et al. [13] and Cook et al. [12].

In girls, fat content, BMI, WC, AVI, BRI and BAIp showed AUC values for MetS above 0.8 and in all cases when the diagnostic was carried out using the IDF criteria. WHR, ABSI and BAI were unable to predict the MetS in girls with a high level of certainty, as AUC values were below 0.8 for all diagnostic criteria employed. In particular, the highest AUC value for ABSI was obtained when the criteria proposed by de Ferranti et al. [15] were used, although it was only 0.590. The lowest AUC values observed corresponded to anthropometric indexes when using the criteria recommended by Weiss et al. [13].

The collinearity test for all anthropometric indexes that included WC for their calculation was found to be negative for WC and AVI (VIF < 3) and positive for, BRI and ABSI (VIF > 3).

## 4. Discussion

In this study, eight different criteria for the diagnosis of MetS in adolescents were used, which resulted in a high variability in prevalence both in boys and girls, but no significant differences in age, fat content or in BMI, which is in agreement with Pergher et al. [8]. In boys, the number of diagnosed individuals ranged from 25 cases when the criteria of Viner et al. [18] were used, up to 68 cases when applying the criteria of Duncan et al. [14], Rodriguez-Moran et al. [17] and Cruz and Goran [16]. This variability could be explained by differences in the cut-off points of the components that make up each of the criteria to define MetS in the adolescent population [32]. There is also a high variability in terms of the components that are considered to be altered for each of the criteria used, which makes it difficult to establish comparisons [33]. Among the criteria used, those proposed by the IDF resulted in the highest average values for the variables weight, WC, HDL cholesterol, SBP, DBP and insulin and the lowest values for glucose, triglycerides, cholesterol and LDL cholesterol. These results differ partially from those obtained in other studies. Sarrafzadegan et al. [34], in their study of an Iranian adolescent population, used the criteria proposed by the IDF and found higher blood pressure levels but lower HDL-c levels compared with our study. On the other hand, Ramírez-Vélez et al. [35], in their study of adolescents from Colombia, also applying the criteria of the IDF, found that the most prevalent altered components were low levels of HDL-c and high levels of triglycerides, whereas the less prevalent components were elevated waist circumference and hyperglycemia. In our study, using the criteria proposed by Viner et al. [18], higher mean levels of glucose, triglycerides, cholesterol, LDL-c and HOMA-IR were obtained. 

In girls, the variability of individuals diagnosed with MetS was even higher, ranging from 18 diagnosed subjects with the criteria of Weiss et al. [13] to 171 with the criteria of Cruz and Goran [16]. According to Nasreddine et al. [36], in their study of adolescents in Lebanon, the differences in the prevalence of MetS among boys and girls could be explained, in part, by the variations in the body composition of the human species, in particular due to a higher fat content in girls. This was also observed in our study, as girls presented a significantly higher fat content, expressed as percentage (Table 2). In addition, in line with Nasreddine et al. [36], the criteria proposed by the IDF showed higher mean values for the variables weight, BMI, WC and SBP in girls.

Regarding the discriminatory capacity for MetS of the anthropometric indexes studied and according to the different diagnostic criteria used, the ROC analysis showed that in boys, only WC, AVI and BRI reached AUC values higher than 0.8, indicating a high predictive capacity. Interestingly, this occurred only when the diagnosis of MetS was performed using the IDF criteria. Consequently, these results show the importance of these indices in the assessment of fat tissue distribution in the body and its relationship with the development of metabolic and cardiovascular disorders at an early age [37].

For the rest of the criteria studied, AUC values lower than 0.8 were obtained, indicating a lower predictive capacity of all anthropometric indexes. Values were particularly low when the criteria used for diagnosis were those proposed by Weiss et al. [13] and Cook et al. [12].

In girls, fat content, BMI, WC, AVI, BRI and BAIp showed values of AUC for the MetS above 0.8 and in all cases when the diagnosis was made using the criteria of the IDF, but not when other criteria were used. On the other hand, WHR, ABSI and BAI showed a lower predictive capacity against MetS in girls, with AUC values below 0.8 for all of the diagnostic criteria used. Based on these findings, it is worth noting the importance of using the IDF criteria for the diagnosis of MetS in adolescents compared to the other criteria in adolescents, in particular when trying to estimate the presence of MetS from anthropometric indexes. These results are in contrast with those obtained by Xu et al. [38], in Chinese male adolescents, who, using the criteria of Cook et al. [12], found AUC values of 0.79 for WHR, higher than those observed in our study. Similarly, Zaki et al. [39], in Egyptian adolescents and using the IDF criteria, found AUC values considerably higher than 0.80 for WHR, while in our study, values were slightly below 0.8 in both boys and girls. These results could suggest a possible influence of factors such as ethnicity and culture on the usefulness of the different criteria to define MetS in adolescents. 

It is noteworthy that, for the ABSI index, the highest value of AUC was obtained only when the criteria proposed by de Ferranti et al. [15] were applied, and was lower than 0.6, indicating a very low predictive capacity. Consequently, and in line with previous studies in adults [40,41] and our previous report in adolescents [6], ABSI should not be used as a predictive index for MetS in adolescents of both genders. As with boys, the lowest values of AUC observed corresponded to the anthropometric indexes after using the criteria recommended by Weiss et al. [13]. At the same time, assuming that WC is used as a component of the MetS, the collinearity test for all the anthropometric indexes that included the WC for calculation showed negative results for WC and AVI (VIF <3) and positive for BRI and ABSI (VIF >3). These results show, once again, that WC and AVI are appropriate anthropometric indicators and of great clinical utility for the diagnosis of MetS in the adolescent population. Cutoff points obtained from ROC analysis for the diagnosis of MetS showed a high variability among the different criteria. Only the cutoff points for WHR and BRI showed a variation coefficient lower than 2% in boys (Appendix A
Appendix A). In girls, variability was even higher, and Fat (%), BMI, BAI, BAIp and BRI showed variation coefficients above 10% (Appendix A). 

The present study has some strengths and limitations. To the best of our knowledge, this study is the first to use eight different criteria for the diagnosis of MetS in adolescents. In addition, we would like to emphasize the usefulness of the large sample size, which allowed us to obtain solid results that are comparable to those of other studies. Furthermore, the acquisition of a representative sample for the age groups contemplated in each region gave this study even greater epidemiological value. Moreover, the fact that all of the adolescents came from the same geographic area and shared the same culture, life style, and nutritional habits increased the homogeneity of sample. This study had various limitations, such as its transversal design, which does not permit causal inference. In addition, there is also a lack of information regarding the puberty status of the participants. For these reasons, the results should be interpreted with caution.

## 5. Conclusions

In conclusion, the results confirm that AVI and WC are the anthropometric indexes that best discriminate between MetS and non-MetS individuals when the criteria proposed by the IDF are used for diagnosis in adolescents. Importantly, the other seven diagnostic criteria were not helpful for this purpose, despite the fact that some of them (Duncan et al. [14] and Cruz and Goran [16], in particular) resulted in a large number of diagnosed individuals, especially in girls. These findings should be considered in future studies and in daily clinical practice, and health professionals should apply the criteria proposed by the IDF. The health authorities should promote the implementation of individual anthropometric indicators in the physical examination that takes place during periodic health checks in the adolescent population. At the same time, new studies with ethnically and culturally different populations are necessary in order to explore in greater depth the usefulness of the different criteria for the diagnosis of MetS and the predictive capacity of all the anthropometric indices studied.

## Figures and Tables

**Figure 1 nutrients-11-01370-f001:**
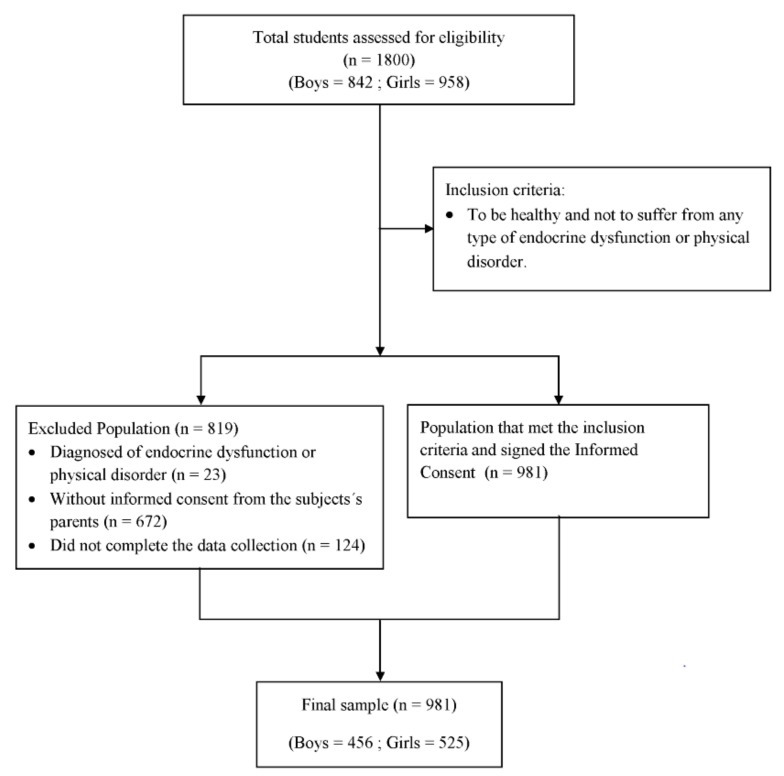
Flow diagram of the recruitment progress.

**Table 1 nutrients-11-01370-t001:** Diagnostic criteria for the MetS in adolescents.

	IDF	Cook	De Ferranti	Weiss	Viner	Duncan	Rodríguez-Moran	Cruz & Goran
Age (years)	10–16	12–19	≥12	4–20	2–18	12–19	10–18	8–13
Number of components	Obesity + 2 components	≥3	≥3	≥3	≥4	≥3	≥3	≥3
Obesity	WC > 90 percentile	WC > 90 percentile	WC > 75 percentile	BMI z-score ≥ 2	BMI ≥ 95 percentile	WC ≥ 90 percentile	WC ≥ 90 percentile	WC ≥ 90 percentile
Glucose (mg/dL)	≥100	≥110	≥110	≥140	≥110	≥110	≥110	<100
TG (mg/dL)	≥150	≥110	≥100	>95 percentile	≥150	≥110	≥90 percentile	≥90 percentile
HDL-cholesterol (mg/dL)	≤40	≤40	<50 girls <45 boys	<5 percentile	≤35	<40	-	<10 percentile
SBP (mmHg)	≥130	>90 percentile	>90 percentile	>95 percentile	>95 percentile	≥90 percentile	≥90 percentile	≥90 percentile
DBP (mmHg)	≥85	-	-	-	-	-	-	-

Notes: IDF, International Diabetes Federation; TG, triglycerides; HDL, high-density lipoprotein; SBP, systolic blood pressure, DBP, diastolic blood pressure, WC, waist circumference; BMI, body mass index.

**Table 2 nutrients-11-01370-t002:** Baseline characteristics of participants.

Variables	Boys (*n* = 456)		Girls (*n* = 525)	
	Mean	SD	Mean	SD
Age (years)	13.2	1.2	13.3	1.2
Weight (kg)	57.1	14.1	53.1	11.0 ***
Fat (%)	27.3	8.3	29.6	7.8 ***
BMI (kg/m^2^)	21.5	4.0	21.1	3.6
WC (cm)	73.7	11.8	71.3	9.6 ***
Glucose (mg/dL)	86.2	31.2	85.2	28.7
TG (mg/dL)	129.2	59.3	125.0	46.2
Cholesterol (mg/dL)	81.8	17.3	81.4	15.7
LDL-c (mg/dL)	93.4	23.6	92.9	22.5
HDL-c (mg/dL)	40.1	2.8	40.0	3.1
SBP (mmHg)	119.6	15.7	116.9	15.1 **
DBP (mmHg)	64.5	9.2	63.9	8.8
Insulin (mU/mL)	21.0	10.2	20.2	9.0
HOMA-IR	4.5	2.9	4.3	3.1

Notes: Fat (%), body fat percentage; BMI, body mass index; WC, waist circumference; TG, triglycerides; LDL-c, low-density lipoprotein cholesterol; HDL-c, high-density lipoprotein cholesterol; SBP, systolic blood pressure, DBP, diastolic blood pressure, HOMA-IR, homeostatic model assessment of insulin resistance. Differences between means were assessed by an unpaired Student’s t-test. **, *p* < 0.05; ***, *p* < 0.001.

**Table 3 nutrients-11-01370-t003:** Characteristics of boys diagnosed of MetS according to the different diagnostic criteria.

Variables	IDF		Cook		De Ferranti		Weiss		Viner		Duncan		Rodríguez-Moran		Cruz & Goran	
	Mean	SD	Mean	SD	Mean	SD	Mean	SD	Mean	SD	Mean	SD	Mean	SD	Mean	SD
MetS (number)	41		39		63		32		25		68		68		68	
Age (y)	13.6 ^a^	1.0	12.9 ^a^	1.2	13.0 ^a^	1.0	13.2 ^a^	1.2	13.0 ^a^	1.2	13.1 ^a^	1.1	13.1 ^a^	1.1	13.1 ^a^	1.1
Weight (kg)	69.3 ^a^	13.6	54.9 ^b^	14.3	59.8 ^b^	15.6	59.3 ^ab^	17.4	59.1 ^ab^	17.0	60.3 ^ab^	15.3	60.3 ^ab^	15.3	60.3 ^ab^	15.3
Fat (%)	34.1 ^a^	6.5	29.3 ^a^	7.8	32.1 ^a^	7.7	29.4 ^a^	8.4	30.3 ^a^	8.1	31.3 ^a^	7.8	31.3 ^a^	7.8	31.3 ^a^	7.8
BMI (kg/m^2^)	25.2 ^a^	4.3	21.8 ^a^	4.2	23.6 ^a^	4.7	22.4 ^a^	5.5	22.4 ^a^	5.0	23.4 ^a^	4.5	23.4 ^a^	4.5	23.4 ^a^	4.5
WC (cm)	86.0 ^a^	9.9	74.6 ^b^	11.7	80.2 ^ab^	12.7	76.9 ^b^	14.0	77.8 ^ab^	13.5	79.5 ^ab^	12.4	79.5 ^ab^	12.4	79.5 ^ab^	12.4
Glucose (mg/dL)	128.4 ^a^	53.5	172.9 ^b^	47.4	139.5 ^a^	56.9	191.4 ^b^	27.8	193.7 ^b^	24.4	136.0 ^a^	56.2	136.0 ^a^	56.2	136.0 ^a^	56.2
TG (mg/dL)	193.0 ^a^	120.1	257.9 ^a^	146.4	210.1 ^ac^	133.9	295.0 ^bc^	143.9	338.9 ^b^	132.5	204.0 ^a^	130.6	204.0 ^a^	130.6	204.0 ^a^	130.6
Chol (mg/dL)	129.6 ^a^	35.6	151.6 ^ab^	27.1	134.6 ^a^	35.7	163.0 ^b^	7.3	163.2 ^b^	8.1	133.4 ^a^	35.4	133.4 ^a^	35.4	133.4 ^a^	35.4
LDL-c (mg/dL)	101.9 ^a^	23.8	119.7 ^b^	24.9	107.2 ^ab^	26.7	126.3 ^c^	19.8	126.0 ^c^	21.3	107.2 ^ab^	25.3	107.2 ^ab^	25.3	107.2 ^ab^	25.3
HDL-c (mg/dL)	35.7 ^a^	3.4	33.1 ^bc^	2.2	35.0 ^a^	3.4	32.3 ^c^	1.6	32.4 ^c^	1.8	34.9 ^ab^	3.2	34.9 ^ab^	3.2	34.9 ^ab^	3.2
SBP (mmHg)	137.2 ^a^	14.7	121.6 ^b^	17.0	123.5 ^b^	17.3	122.9 ^b^	17.9	123.4 ^b^	18.8	124.0 ^b^	16.6	124.0 ^b^	16.6	124.0 ^b^	16.6
DBP mmHg)	74.1 ^a^	11.4	64.6 ^b^	8.6	65.5 ^b^	9.0	65.1 ^b^	9.0	63.9 ^b^	9.2	66.3 ^b^	8.7	66.3 ^b^	8.7	66.3 ^b^	8.7
Insulin (mU/mL)	28.7 ^a^	15.1	20.9 ^a^	10.4	25.5 ^a^	13.6	23.7 ^a^	14.2	24.0 ^a^	14.3	24.9 ^a^	13.3	24.9 ^a^	13.3	24.9 ^a^	13.3
HOMA-IR	8.8 ^a^	6.0	9.1 ^a^	5.7	8.3 ^a^	5.1	10.7 ^a^	5.6	11.1 ^a^	6.2	8.0 ^a^	5.0	8.0 ^a^	5.0	8.0 ^a^	5.0

Notes: Fat (%), body fat percentage; BMI, body mass index; WC, waist circumference; TG, triglycerides; Chol, cholesterol; LDL-c, low-density lipoprotein cholesterol; HDL-c, high-density lipoprotein cholesterol; SBP, systolic blood pressure, DBP, diastolic blood pressure, HOMA-IR, homeostatic model assessment of insulin resistance. Differences between means that share a letter are not statistically significant (*p* < 0.05).

**Table 4 nutrients-11-01370-t004:** Characteristics of girls diagnosed of MetS according to the different diagnostic criteria.

Variables	IDF		Cook		de Ferranti		Weiss		Viner		Duncan		Rodríguez-Moran		Cruz & Goran	
	Mean	SD	Mean	SD	Mean	SD	Mean	SD	Mean	SD	Mean	SD	Mean	SD	Mean	SD
MetS (number)	32		58		97		18		21		134		86		171	
Age (years)	13.3 ^a^	1.1	12.9 ^a^	1.1	12.9 ^a^	1.1	13.1 ^a^	1.4	13.2 ^a^	1.3	13.0 ^a^	1.1	13.1 ^a^	1.1	13.1 ^a^	1.1
Weight (kg)	67.1 ^a^	10.1	51.8 ^b^	12.8	55.4 ^bc^	11.9	57.0 ^ab^	18.0	59.2 ^ab^	16.6	54.1 ^b^	10.4	60.3 ^ac^	15.3	54.8 ^b^	11.0
Fat (%)	38.1 ^a^	6.6	30.0 ^a^	8.4	32.6 ^a^	8.0	31.9 ^a^	11.3	33.0 ^a^	9.9	31.9 ^a^	7.5	31.3 ^a^	7.8	32.0 ^a^	7.6
BMI (kg/m^2^)	25.8 ^a^	3.5	21.1 ^b^	4.3	22.5 ^bc^	4.0	22.4 ^ab^	5.9	23.1 ^ab^	5.2	22.1 ^bc^	3.6	23.4 ^a^	4.5	22.1 ^bc^	3.7
WC (cm)	83.2 ^a^	7.5	70.5 ^b^	10.4	75.2 ^bc^	10.3	73.7 ^bc^	14.8	75.5 ^ab^	13.3	73.6 ^b^	9.1	79.5 ^ac^	12.4	73.7 ^b^	9.5
Glucose (mg/dL)	132.0 ^abc^	52.2	148.3 ^b^	51.8	121.1 ^a^	52.3	194.4 ^d^	7.1	188.2 ^d^	22.5	110.0 ^c^	48.2	136.0 ^ab^	56.2	104.5	43.9 ^c^
TG (mg/dL)	196.2 ^ab^	120.9	191.6 ^ab^	114.9	164.5 ^ab^	98.1	329.8 ^c^	121.9	312.9 ^c^	128.1	151.9 ^a^	85.9	204.0 ^b^	130.6	145.5 ^a^	77.0
Chol (mg/dL)	132.3 ^ab^	33.2	139.9 ^a^	30.9	122.1 ^b^	35.8	161.4 ^a^	4.8	160.0 ^a^	4.9	114.1 ^c^	34.8	133.4 ^a^	35.4	109.3 ^c^	32.5
LDL-c (mg/dL)	106.3 ^ab^	20.0	109.3 ^a^	21.2	98.2 ^b^	23.3	126.1 ^a^	16.1	122.8 ^a^	15.9	92.9 ^c^	22.3	107.2 ^a^	25.3	90.1 ^c^	21.2
HDL-c (mg/dL)	35.3 ^ab^	2.6	33.7 ^a^	4.5	36.1 ^b^	4.8	32.8 ^a^	1.3	33.1 ^a^	1.3	37.2 ^c^	4.6	34.9	3.2 ^a^	38.0	4.3 ^c^
SBP (mmHg)	132.3 ^a^	10.2	113.4 ^b^	17.1	117.7 ^bc^	17.0	116.4 ^bc^	20.8	118.8 ^ab^	19.5	117.2 ^b^	15.5	124.0 ^ac^	16.6	117.5 ^b^	15.3
DBP mmHg)	70.7 ^a^	8.8	62.1 ^a^	8.8	62.6 ^a^	9.5	64.4 ^a^	10.2	64.4 ^a^	10.1	63.0 ^a^	9.0	66.3 ^a^	8.7	63.7 ^a^	8.9
Insulin (mU/mL)	11.0 ^a^	8.0	8.4 ^b^	6.6	7.1 ^bc^	5.6	13.8 ^a^	8.7	13.6 ^a^	8.5	6.1 ^b^	5.0	8.0 ^ab^	5.0	5.8 ^c^	4.6
HOMA-IR	32.5 ^a^	15.8	22.0 ^b^	12.2	23.6 ^b^	12.3	28.8 ^ab^	17.9	28.8 ^ab^	17.1	21.7 ^b^	10.4	24.9 ^ab^	13.3	21.8 ^b^	10.5

Notes: Fat (%), body fat percentage; BMI, body mass index; WC, waist circumference; TG, triglycerides; Chol, cholesterol; LDL-c, low-density lipoprotein cholesterol; HDL-c, high-density lipoprotein cholesterol; SBP, systolic blood pressure, DBP, diastolic blood pressure, HOMA-IR, homeostatic model assessment of insulin resistance. Differences between means that share a letter are not statistically significant (*p* < 0.05).

**Table 5 nutrients-11-01370-t005:** Area under the curve (AUC) in receiver-operator curve (ROC) analysis of different anthropometric indexes for predicting MetS components in boys according to diagnostic criteria.

Variables	IDF	Cook	De Ferranti	Weiss	Viner	Duncan	Rodríguez-Moran	Cruz & Goran	Maximum	Author	Minimum	Author
Fat (%)	0.757	0.582	0.696	0.576	0.612	0.666	0.666	0.666	0.757	IDF	0.576	Weiss
BMI	0.783	0.523	0.663	0.523	0.542	0.657	0.657	0.657	0.783	IDF	0.523	Weiss and Cook
WC	0.831	0.526	0.680	0.564	0.601	0.669	0.669	0.669	0.831	IDF	0.526	Cook
WHR	0.789	0.609	0.715	0.635	0.690	0.655	0.655	0.655	0.789	IDF	0.609	Cook
ABSI	0.663	0.559	0.652	0.564	0.617	0.615	0.615	0.615	0.663	IDF	0.559	Cook
BAI	0.686	0.559	0.689	0.496	0.512	0.664	0.664	0.664	0.689	de Ferranti	0.496	Weiss
AVI	0.831	0.524	0.678	0.562	0.599	0.668	0.668	0.668	0.831	IDF	0.524	Cook
BRI	0.800	0.590	0.728	0.572	0.615	0.700	0.700	0.700	0.800	IDF	0.572	Weiss
CI	0.767	0.577	0.706	0.593	0.644	0.675	0.675	0.675	0.767	IDF	0.577	Cook
BAIp	0.752	0.521	0.671	0.499	0.515	0.668	0.668	0.668	0.752	IDF	0.499	Weiss

Notes: ABSI, a body shape index; AVI, abdominal volume index; BAI, body adiposity index; BAIp, pediatric body adiposity index; BMI, body mass index; BRI, body roundness index; C-Index, conicity index; Fat (%), body fat percentage; WC, waist circumference; WHR, waist-to-hip ratio. Maximum and minimum indicate the highest and lowest AUC values observed, together with the corresponding author of the diagnostic criteria.

**Table 6 nutrients-11-01370-t006:** Area under the curve (AUC) in receiver-operator curve (ROC) analysis of different anthropometric indexes for predicting MetS components in girls according to diagnostic criteria.

Variables	IDF	Cook	De Ferranti	Weiss	Viner	Duncan	Rodríguez-Moran	Cruz & Goran	Maximum	Author	Minimum	Author
Fat (%)	0.812	0.638	0.638	0.559	0.605	0.619	0.557	0.634	0.812	IDF	0.557	Rodriguez-Moran
BMI	0.855	0.635	0.635	0.517	0.607	0.627	0.557	0.636	0.855	IDF	0.517	Weiss
WC	0.866	0.645	0.645	0.535	0.614	0.616	0.554	0.626	0.866	IDF	0.535	Weiss
WHR	0.717	0.662	0.662	0.528	0.557	0.536	0.533	0.537	0.717	IDF	0.528	Weiss
ABSI	0.585	0.590	0.590	0.489	0.496	0.541	0.511	0.539	0.590	de Ferranti	0.489	Weiss
BAI	0.797	0.651	0.651	0.513	0.585	0.714	0.590	0.702	0.797	IDF	0.513	Weiss
AVI	0.867	0.643	0.643	0.535	0.616	0.619	0.554	0.629	0.867	IDF	0.535	Weiss
BRI	0.848	0.677	0.677	0.519	0.594	0.663	0.575	0.661	0.848	IDF	0.519	Weiss
CI	0.730	0.642	0.642	0.520	0.552	0.597	0.542	0.593	0.730	IDF	0.520	Weiss
BAIp	0.840	0.624	0.624	0.525	0.608	0.687	0.575	0.688	0.840	IDF	0.525	Weiss

Notes: ABSI, a body shape index; AVI, abdominal volume index; BAI, body adiposity index; BAIp, pediatric body adiposity index; BMI, body mass index; BRI, body roundness index; C-Index, conicity index; Fat (%), body fat percentage; WC, waist circumference; WHR, waist-to-hip ratio. Maximum and minimum indicate the highest and lowest AUC values observed, together with the corresponding author of the diagnostic criteria.

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
