# Peer review of "Waist Circumference and Abdominal Volume Index Can Predict Metabolic Syndrome in Adolescents, but only When the Criteria of the International Diabetes Federation are Employed for the Diagnosis"

_nutrients, 2019, doi:10.3390/nu11061370_

Reviewer 1 Report

This paper reports the results of an interesting study aimed to show that waist circumference and abdominal volume index were able, alone, to predict metabolic syndrome (MetS) in adolescents. 

Obesity-related comorbidities, also in pediatric population, are ones of the most relevant medical issue nowadays. MetS is an excellent estimator of these comorbidities, for this reason, this is a relevant study and could improve the readers insight about the debated criteria to diagnose MetS.

This paper reports results on a large adolescent population. This is an important strength both because it represents a large effort and statistical solid results and because this is a complex and understudied pediatric population.

Although in my humble opinion the paper could be of interest for the scientific readers as it currently is, I suggest improving it according to the following consideration:

-The variability reported in the girls’ subgroup of the studied population is higher than in the boys' one. According to the early puberty age of the girls compared to the boys', this result could suggest an important role of the puberty stage in the fat distribution and so in the MetS criteria correctness. 

An evaluation of the puberty stage, in particular in girls’ subgroup, with a statistical evaluation and stratification could improve the definition suggestinf ad hoc criteria in this delicate age.

Minor revision:

Please define the acronyms at the first time they appear.  

Author Response

Comments and Suggestions for Authors

This paper reports the results of an interesting study aimed to show that waist circumference and abdominal volume index were able, alone, to predict metabolic syndrome (MetS) in adolescents. Obesity-related comorbidities, also in pediatric population, are ones of the most relevant medical issue nowadays. MetS is an excellent estimator of these comorbidities, for this reason, this is a relevant study and could improve the readers insight about the debated criteria to diagnose MetS. This paper reports results on a large adolescent population. This is an important strength both because it represents a large effort and statistical solid results and because this is a complex and understudied pediatric population.

Although in my humble opinion the paper could be of interest for the scientific readers as it currently is, I suggest improving it according to the following consideration:

- The variability reported in the girls’ subgroup of the studied population is higher than in the boys' one. According to the early puberty age of the girls compared to the boys', this result could suggest an important role of the puberty stage in the fat distribution and so in the MetS criteria correctness. An evaluation of the puberty stage, in particular in girls’ subgroup, with a statistical evaluation and stratification could improve the definition suggesting ad hoc criteria in this delicate age.

Response: The reviewer’s suggestion is extremely relevant but unfortunately, we do not possess this information. This problem has thus been included at the end of the discussion as one of the limitations of the research study.

Minor revision:

- Please define the acronyms at the first time they appear. 

Response: As suggested by the reviewer, acronyms have been revised and defined when they first appear.

Reviewer 2 Report

In this work, Perona et al. have deal with an important problem in the medical practice: diagnosing of metabolic syndrome in adolescents.

As it is known, obesity and related pathologies are always greater in young people metabolic syndrome’s diagnosis is not easy considering the different evolutionary processes. The authors have proposed an interesting study in order to identify simple parameters (primary anthropometric parameters waist circumference and abdominal volume index) that can allow the diagnosis of metabolic syndrome in adolescents.

Despite the limitations in the study, which the authors themselves emphasize in the final part of the discussion, the work is well written, it is useful for authors that approach the problem by defining few  parameters useful for the diagnosis of metabolic syndrome in adolescents. The cross-sectional study design is adequate and the statistical analysis is appropriate. Data obtained highlight the problems linked to the sex of the patients. Furthermore, the discussion offers interesting points for reflection.

If possible, I would advise the authors to add very few lines in introduction section in order to highlight even more the need to identify a metabolic syndrome diagnosis system for adolescents that is as simple and easy as possible and the solution of this problem is urgent considering the exponential increase of the pathology in this age group.

Author Response

Comments and Suggestions for Authors

In this work, Perona et al. have deal with an important problem in the medical practice: diagnosing of metabolic syndrome in adolescents. As it is known, obesity and related pathologies are always greater in young people metabolic syndrome’s diagnosis is not easy considering the different evolutionary processes. The authors have proposed an interesting study in order to identify simple parameters (primary anthropometric parameters waist circumference and abdominal volume index) that can allow the diagnosis of metabolic syndrome in adolescents. Despite the limitations in the study, which the authors themselves emphasize in the final part of the discussion, the work is well written, it is useful for authors that approach the problem by defining few parameters useful for the diagnosis of metabolic syndrome in adolescents. The cross-sectional study design is adequate and the statistical analysis is appropriate. Data obtained highlight the problems linked to the sex of the patients. Furthermore, the discussion offers interesting points for reflection.

- If possible, I would advise the authors to add very few lines in introduction section in order to highlight even more the need to identify a metabolic syndrome diagnosis system for adolescents that is as simple and easy as possible and the solution of this problem is urgent considering the exponential increase of the pathology in this age group.

Response: As requested by the reviewer, that information has been included in introduction.